# Neural Geometric Fabrics: Efficiently Learning High-Dimensional Policies from Demonstrations

**Mandy Xie**[1,2],[*] **Ankur Handa**[1], **Stephen Tyree**[1], **Dieter Fox**[1,3],
**Harish Ravichandar**[2], **Nathan Ratliff**[1], **Karl Van Wyk**[1]
[1] NVIDIA, [2] Georgia Institute of Technology, [3] University of Washington.

**Abstract:** Learning dexterous manipulation policies for multi-fingered robots has been a long-standing challenge in robotics. Existing methods either limit themselves to highly constrained problems and smaller models to achieve extreme sample efficiency or sacrifice sample efficiency to gain capacity to solve more complex tasks with deep neural networks. In this work, we develop a *structured* approach to sample-efficient learning of dexterous manipulation skills from demonstrations by leveraging recent advances in robot motion generation and control. Specifically, our policy structure is induced by Geometric Fabrics - a recent framework that generalizes classical mechanical systems to allow for flexible design of expressive robot motions. To avoid the cumbersome manual design required by existing motion generators, we introduce *Neural Geometric Fabric (NGF)* - a framework that learns Geometric Fabric-based policies from data. NGF policies are provably stable and capable of encoding speed-invariant geometries of complex motions in multiple task spaces simultaneously. We demonstrate that NGFs can learn to perform a variety of dexterous manipulation tasks on a 23-DoF hand-arm physical robotic platform purely from demonstrations. Results from comprehensive comparative and ablative experiments show that NGF's structure and action spaces help learn acceleration-based policies that consistently outperform state-of-the-art baselines like Riemannian Motion Policies (RMPs), and other commonly used networks, such as feed-forward and recurrent neural networks. More importantly, we demonstrate that NGFs do *not* rely on often-used and expertly-designed operational-space controllers, promoting an advancement towards efficiently learning safe, stable, and high-dimensional controllers.

**Keywords:** Imitation Learning, Dexterous Manipulation

## 1 Introduction

Autonomous and robust dexterous manipulation capabilities could enable robots to perform a wide range of tasks, such as opening a door and using tools, in a world built by and for humans. However, dexterous manipulation involving multi-finger hands continues to be a long-standing challenge in robotics, with human-level dexterity remaining a distant target. Indeed, there are numerous factors contributing to this state of affairs, such as complex dynamics, high-dimensional action spaces, and the difficulties involved in designing bespoke controllers that do not generalize to either novel tasks or variations of the same task, such as new object poses or different targets.

Recent efforts in robot learning have aimed to study and tackle the challenges that plague autonomous dexterous manipulation [1–11]. Specifically, recent learning-based techniques show promise by generating appropriate robot behavior for a variety of tasks without the need for hand-crafted motion policies. Most existing approaches can be divided into two broad areas: i) methods that utilize deep neural networks to learn highly parameterized, high-dimensional policies that solve complex dexterous manipulation tasks with minimal manual design, but at the cost of low sample efficiency and lack of theoretical guarantees [6–8, 12–15], and ii) methods that utilize highly structured models with significantly fewer parameters that can achieve extreme data efficiency, but are either limited to highly constrained problems or depend on expertly-designed operational space controllers [16–26].

In this work, we attempt to combine the best of both high-capacity models and sample-efficient structures in order to learn high-dimensional dexterous manipulation policies for a *physical* hand-

---

[*]manxie@gatech.edu

6th Conference on Robot Learning (CoRL 2022), Auckland, New Zealand.

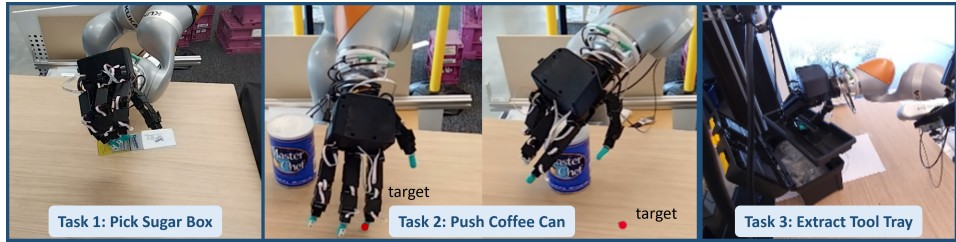

Figure 1: NGFs can learn to control all the joints of a hand-arm system to perform tasks, such as (a) grasping and lifting a box, (b) touching a target point before pushing a can over it, and (c) extracting a toolbox tray.

arm system purely from a limited number of demonstrations. We are concerned with learning policies that directly and simultaneously coordinate all the *joint-space* degrees of freedom of both the arm and the hand and can generate stable and smooth motions for tasks requiring dexterity, such as grasping and manipulating an object with a multi-fingered robotic hand.

Inspired by the growing evidence in support of structured policies for manipulation (e.g., [27, 28]), we propose an algorithm to learn highly-structured and generalizable policies for dexterous manipulation from a limited number of demonstrations. Specifically, we leverage the structure imposed by *Geometric Fabrics* [29], a recent framework that generalizes classical geometric control (e.g., operational space control) using tools from differential geometry, to facilitate stable and flexible robot motion generation [30]. Geometric fabrics are stable second-order dynamical systems, which encode behaviors as speed-invariant paths (via geometries) and generate acceleration actions to generate intended motions. Further, learning to encode dynamical systems via acceleration is known to be a significantly beneficial form of inductive bias in policy learning [31]. However, designing these fabrics for complex dexterous manipulation tasks requires considerable expertise and effort.

To circumvent the need for significant expertise and painstaking human effort in designing Geometric Fabrics for dexterous manipulation tasks, we introduce *Neural Geometric Fabrics* (NGF), a framework to efficiently learn components of Geometric Fabrics, such as subtask maps, geometries, and priority metrics. We discuss details of our architecture and learning pipeline in Section 4.

The primary contributions of this paper are: 1) a novel framework to learn Geometric-Fabrics-based policies directly from demonstrations, removing the need for manual design and expertise; 2) a thorough investigation of the benefits of additional structure imposed by Geometric Fabrics over existing approaches when learning dexterous manipulation skills from demonstrations; and 3) to the best of our knowledge, this is the first framework to train and deploy policies that directly control all joints of the highly-actuated arm-hand system in the real world without relying on manually-designed operational-space controllers in elevated actions spaces.

We evaluate the effectiveness of our approach on a 23-DoF physical hand-arm robot platform in the real world across three different tasks (see Fig 1), and compare its performance against several baselines inspired by the state-of-the-art in learning from demonstration. Our results show that the policies learned with our structured approach consistently outperform baseline methods with respect to metrics measuring task success, safety of deployment, and sample efficiency.

## 2  Related Work

A variety of approaches have attempted to tackle the challenges of learning dexterous manipulation polices. Here, we discuss how our approach relates to a few different categories of existing methods.

**Deep neural network based approaches** such as deep Reinforcement Learning (RL) methods [7–9, 14, 32] have demonstrated impressive capabilities. However, most RL algorithms suffer from poor sample efficiency. On the other hand, Imitation Learning (IL) improves sample efficiency by leveraging expert demonstrations [1, 33]. However, pure IL methods are often limited to non-dexterous manipulation [2, 26, 34, 35]. Recent approaches have combined IL with RL in efforts to improve data efficiency in learning complex dexterous manipulation skills [3–6, 12, 36]. However, these methods are often limited to simulation as collecting exploratory data can be expensive, time-intensive, and potentially damaging to the hardware in real-world robotic systems. Given these

observations, it is critical to develop robot learning frameworks that can learn to generalize from less data. We demonstrate that NGF can learn complex dexterous manipulation skills directly on hardware purely via imitation, without expertly-designed operational space controllers.

**Exploiting structures** in robotic manipulation problems has shown to improve data efficiency in policy learning even though structured policy parameterizations are more restricted in representation power than their non-structured counterparts [37]. This is due to the fact that, if the restrictions provide appropriate inductive biases for the task of interest, structured approaches provide numerous advantages, such as provable stability, safety guarantees, and improved efficiency, without adversely impacting expressivity. Indeed, there are several successful examples of learning structured policies. The approaches in [38–42] leverage structures from control theory, where learned policies meet optimal control assumptions to provide stability guarantees. Dynamical system based approaches [16–26] learn policies by imposing structures or constraints that enforce stability and convergence while imitating the demonstrations. Recently, it was shown that *Riemannian Motion Policies (RMPs)* [27, 28, 43, 44] can encode the geometric structure of manipulation tasks using structured second-order dynamical systems. One can view the proposed NGFs as specialized versions of RMPs that are stable and capable of encoding more complex behaviors as geometries, providing a more appropriate inductive bias for dexterous manipulation. See Appendix L for a more detailed comparison between NGF and RMP.

**The choice of action spaces** plays a significant role in learning manipulation policies [37], with the most popular choice being the end-effector pose space [2, 26, 32, 33] as it is more directly relevant to the task and can generalize across different robot kinematics [45, 46]. These approaches rely on an operational space controller to map these desired signals into the joint space. However, it is not trivial to design such a controller for highly kinematically-redundant robots, such as multi-fingered hands. In contrast, our method does not require any hand-crafted operational-space controllers as it leverages the robot's kinematic model to directly learn in multiple actions spaces simultaneously.

## 3 Background: Geometric Fabrics

Geometric fabrics are the latest incarnation of research in robot motion generation and control as defined by second order equations. Geometric fabrics build upon the more general, intuitive, and flexible aspects of Rimennian Motion Policies (RMPs), but retain the stability guarantees offered by classical mechanical systems (which are often leveraged in control as geometric or operational-space control.) In fact, Geometric Fabrics generalize classical mechanical systems and have been shown to outperform RMPs [29]. Geometric fabrics provide a formal mathematical framework for building control policies and primarily encode behavior as a set of nonlinear *geometries of paths*. This geometrically consistent behavior is known as the nominal behavior of the system and acts as an optimization medium over which additional policies derived from potential functions and damping can be applied to further shift the resulting behavior. Policies of this design are guaranteed to be stable, i.e., the system comes to rest at the minimum of the net potential acting on the system.

### 3.1 Nonlinear Geometries

A generalized nonlinear geometry is an acceleration policy $\ddot{\mathbf{x}} + \pi(\mathbf{x}, \dot{\mathbf{x}}) = \mathbf{0}$ where $\pi$ is *homogeneous of degree* 2 (HD2), meaning for any $\lambda \geq 0$ we have $\pi(\mathbf{x}, \lambda\dot{\mathbf{x}}) = \lambda^2 \pi(\mathbf{x}, \dot{\mathbf{x}})$. The HD2 property ensures that every integral curve starting from a given task-space position $\mathbf{x}_0$ with velocity $\dot{\mathbf{x}}_0 = \eta\widehat{\mathbf{n}}$ will follow the same path. The geometric consistency property turns $\pi$ into a *geometry of paths*. From a imitation learning perspective, this additional structure promotes learning speed-invariant *paths* through space instead of *trajectories*, which are sensitive to the speed of travel. We postulate the added structure will be beneficial for sample-efficiency in policy learning.

### 3.2 Geometric Fabrics: encode nominal behaviors

A Geometric Fabric is a system that evolves according to $\mathbf{M}(\mathbf{q}, \dot{\mathbf{q}})\ddot{\mathbf{q}} + \mathbf{f}(\mathbf{q}, \dot{\mathbf{q}}) = \mathbf{0}$ in root coordinates $\mathbf{q}$, where $\ddot{\mathbf{q}} = -\mathbf{M}^{-1}\mathbf{f}$ is a nonlinear geometry that defines a nominal behavior and $\mathbf{M}$ is the priority metric (we drop function variables for notational simplicity). As discussed in Theorem IV.5 of [29], we can energize this Geometric Fabric to conserve energy: $\ddot{\mathbf{q}} = -\mathbf{P}_e[\mathbf{M}^{-1}\mathbf{f}]$, where $\mathbf{P}_e = \mathbf{M}^{\frac{1}{2}}\left[\mathbf{I} - \hat{\mathbf{v}}\hat{\mathbf{v}}^T\right]\mathbf{M}^{-\frac{1}{2}}$ is a metric-weighted projection matrix with $\mathbf{v} = \mathbf{M}^{\frac{1}{2}}\dot{\mathbf{q}}$ and $\hat{\mathbf{v}} = \mathbf{v}/\|\mathbf{v}\|$,

which projects acceleration to the orthogonal space of velocity, hence conserves energy by performing no work. The energized fabric defines a nominal behavior independent of a specific task.

## 3.3 Optimize a potential: solve a task

The nominal behavior can be pushed away from by a potential function to achieve task goals, and damping dissipation ensures convergence. The desired equation of motion of the system then becomes: $\ddot{\mathbf{q}}_d = -\mathbf{P}_e\left[\mathbf{M}^{-1}\mathbf{f}\right] - \mathbf{M}^{-1}\partial_{\mathbf{q}}\psi(\mathbf{q}) - \mathbf{B}\dot{\mathbf{q}}$, where $\psi(\mathbf{q})$ is a potential function, and $\mathbf{B}$ is a positive definite damping matrix. The potential function induces kinetic energy, resulting in motion, and together with damping, drives the system towards the potential's local minima, where motion ultimately comes to rest at the minima of the potential function, hence its optimization.

## 3.4 Policy Design with Geometric Fabrics

For robot manipulators, a desired behavior may involve coordinated motion of different parts. Geometries can be added to spaces of a transform tree [47] for the modular design of composite behaviors. We can construct a transform tree with root node in the configuration space $\mathcal{C}$ (typically joint space) of the robot, and add a leaf node in each subtask space $\mathcal{T}_k$ in which the task is more convenient to be described. For instance, a goal-reaching task can be defined in the 3-D Euclidean space describing the position of the end-effector in relation to the goal. We denote subtask spaces $\{\mathcal{T}_k\}_{k=1}^{K}$, where $K$ is the number of subtasks. Let $\phi_k : \mathcal{C} \to \mathcal{T}_k$ be the mapping from the configuration space $\mathcal{C}$ to subtask space $\mathcal{T}_k$. An example of $\phi_k$ is the forward kinematic mapping, where the subtask space $\mathcal{T}_k$ is the end-effector frame. We then populate each node with a priority metric and geometry pair, $[\mathbf{M}_k, \pi_k]$, which are pulled-back and combined at the root, resulting in a desired configuration space policy $\ddot{\mathbf{q}}_d = \pi_{\mathcal{C}}(\mathbf{q}, \dot{\mathbf{q}})$. In the case where we only have an end-effector space, solving for $\ddot{\mathbf{q}}_d$ is equivalent to a pseudo-inverse as described in [48]. We include additional details about composing policies from sub-task spaces in Appendix A.

# 4 Neural Geometric Fabrics

In this section, we introduce our NGF policy architecture and its learnable components. We illustrate our design process in Fig. 2, and elaborate on each step in the sections below.

## 4.1 Define subtask spaces

We begin by defining two subtask spaces: (1) A 3-D Euclidean space describing the position of the palm in relation to the object, which is stacked with an Eigenspace of the hand. This map is defined as $\phi_1 : \mathbf{q} \mapsto [\mathtt{fk}^T(\mathbf{q}) - \mathbf{x}_o^T, \ \mathtt{pca}^T(\mathbf{q})]^T \in \mathbb{R}^{8\times 1}$, where $\mathtt{fk}(\mathbf{q})$ is the forward kinematics mapping to a palm point and $\mathbf{x}_o$ is the object position. $\mathtt{pca}(\mathbf{q})$ is a linear map of the hand, inspired by the concept of Eigengrasp [49], where we first apply Principle Component Analysis (PCA) to the 16-DoF hand joints data, which reduces the 16-DoF joint space to the 5-DoF Eigenspace while the first five principle components account for more than 95% of the variance. This stacked space directly shares information between the arm and hand, allowing them to coordinate properly. (2) The full configuration space of the robot, $\phi_2 : \mathbf{q} \mapsto \mathbf{q} \in \mathbb{R}^{23\times 1}$. The policy in this space additionally shapes the configuration space behavior, filling in the nullspace of the preceding space.

## 4.2 Define learnable components in subtask spaces

**Geometric policies:** A priority metric and geometry pair in the stacked space $\phi_1$ is defined as $[\mathbf{M}_1, \pi_1]$, encoding object-reaching and coordinated palm-finger behavior. Another priority metric and geometry pair $[\mathbf{M}_2, \pi_2]$ is defined in the configuration space, $\phi_2$, that serves as a residual policy, shaping the configuration space behavior and ensuring the combined Geometric Fabric is full rank.

**Potential and damping:** We additionally add another priority metric and acceleration-based potential policy $[\mathbf{M}_f, \pi_f]$ ($\pi_f$ is purely a function of position) in the stacked space $\phi_1$ to: 1) maintain bounded system energy levels while 2) induce motion (energized geometries cannot induce motion by themselves when at rest), and 3) optimize the corresponding potential function. Finally, a strictly positive damping scalar $b$ in the configuration space, $\phi_2$, is also learned and applied that removes energy from the system, ensuring convergence to the minima of the potential function.

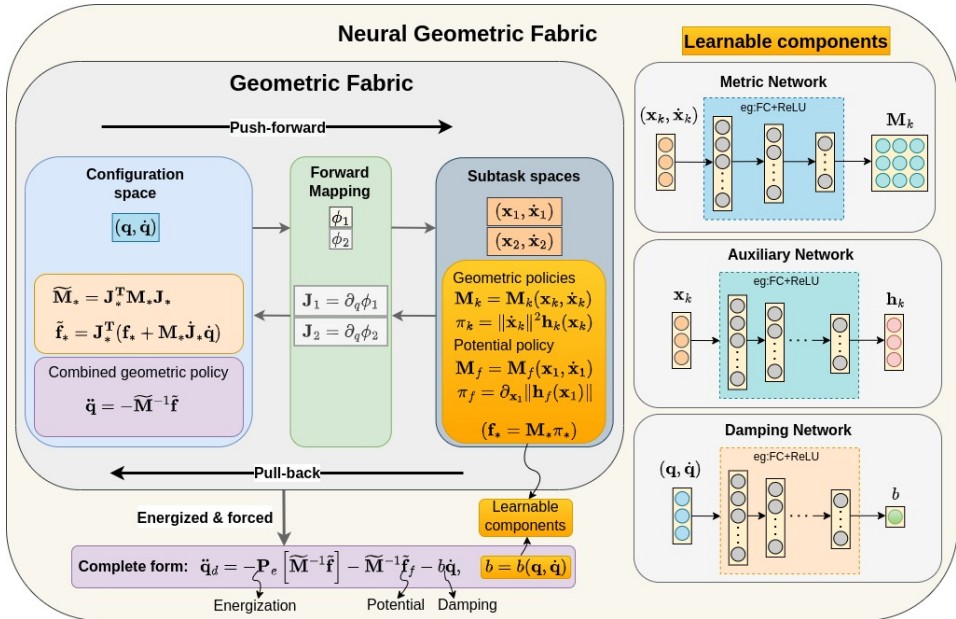

Figure 2: We parameterize NGFs with neural networks and construct them as follows: (1) define each subtask space with a forward mapping $\phi_k$ that maps coordinates $\mathbf{q}$ in the configuration space into coordinates $\mathbf{x}_k$ in the k-th subtask space, (2) define a geometric policy pair $[\mathbf{M}_k, \pi_k]$ in each subtask space $\mathbf{x}_k$, e.g., $k = 1, 2$, and define a potential policy pair $[\mathbf{M}_f, \pi_f]$ in the desired sub-task space, e.g., $\mathbf{x}_1$, (3) pull-back the geometric and potential policy pairs into the configuration space, (4) combine the geometric policies via a metric-weighted average, (5) energize the combined policy (project orthogonal to the direction of motion with $\mathbf{P}_e$) to create a collection of energy-preserving paths (the Geometric Fabric), and (6) force the Geometric Fabric with a potential defined by $[\mathbf{M}_f, \pi_f]$ and damp via $b$ applied along $\dot{\mathbf{q}}$, which ensures convergence to the potential's minima. Note that we parameterize the geometric policy pairs $[\mathbf{M}_k, \pi_k]$, the potential policy pair $[\mathbf{M}_f, \pi_f]$, and the damping scalar $b$ with network networks and learn them from data. We can also define $\phi_k$ with neural networks if desired. For notation simplicity, we use subscript $*$ to represent $1, 2$, and $f$.

All learnable components, including $[\mathbf{M}_*, \pi_*]$ ($\pi_*$ is constructed with $\mathbf{h}_*$ and $* = 1, 2, f$) and the damping scalar $b$, are defined with neural networks. To ensure the priority metrics are positive definite, we follow the architecture described in [28]. See Appendix B for detailed parameterizations.

## 4.3 Complete form

Using the pullback and combination operations as defined in Appendix A, the two geometric policies and potential-based policy are combined at the root along with damping and energization. With the total metric, $\widetilde{\mathbf{M}} = \widetilde{\mathbf{M}}_1 + \widetilde{\mathbf{M}}_2 + \widetilde{\mathbf{M}}_f$, total geometric force, $\widetilde{\mathbf{f}} = \widetilde{\mathbf{f}}_1 + \widetilde{\mathbf{f}}_2$, and potential force $\widetilde{\mathbf{f}}_f$, the desired NGF acceleration takes the form $\ddot{\mathbf{q}}_d = -\mathbf{P}_e\left[\widetilde{\mathbf{M}}^{-1}\widetilde{\mathbf{f}}\right] - \widetilde{\mathbf{M}}^{-1}\widetilde{\mathbf{f}}_f - b\dot{\mathbf{q}}$, where $\mathbf{P}_e = (\mathbf{I} - \hat{\mathbf{q}}\hat{\mathbf{q}}^T)$ energizes the fabric with a squared velocity norm energy (see equation (10) of [29] concerning this energization), and $b \in \mathbb{R}^+$ is a damper. $\ddot{\mathbf{q}}_d$ is time integrated to produce $\mathbf{q}_d$ and $\dot{\mathbf{q}}_d$, the *command* trajectory for the robot to follow (see algorithms in Appendix C).

# 5  Learning Policies from Demonstrations

Given the policy architecture in Section 4, here we describe how we train NGF-based policies using demonstrations to solve manipulation tasks on a 23-DoF hand-arm system system (7-DoF arm and a 16-DoF multi-fingered hand). We used the DexPilot teleoperation system [50] to collect task-solving demonstrations, and trained a policy to replicate the demonstrated joint position trajectories.

## 5.1 Problem Statement

Consider a deterministic discrete-time system with transition model $s_{t+1} = f(s_t, a_t)$ where $s_t \in \mathbb{R}^n$ is the state, and $a_t \in \mathbb{R}^m$ is the action of the system, and $f : \mathbb{R}^n \times \mathbb{R}^m \to \mathbb{R}^n$ is the system transition function. We want to learn an NGF policy $\pi_\theta : s \mapsto \pi_\theta(s)$, $\Pi := \{\pi_\theta : \theta \in \Theta\}$, given $N$ trajectory demonstrations $\{\tau^{(i)}\}_{i=1}^N$. Each demonstration is defined as a sequence of states, denoted $\tau^{(i)} := \{s_t^{(i)}\}_{t=0}^{T_i}$, where $T_i$ is the horizon for the $i$th demonstration. Note, the system transition model we refer to here is simply Euler integration defined by Eq. (7) in Appendix D.

## 5.2 Policy Optimization

To mitigate the distribution shift issue involved in behavior cloning, we formulate the policy learning problem as a multi-step prediction error optimization problem, in which we learn a policy that can reproduce the demonstrated behavior by minimizing the deviation between the demonstrated trajectory and trajectory produced by a learned policy under the system transition function $f$. That is, given a transition function $s_{t+1} := f(s_t, \pi_\theta(s_t))$ for each policy, we can rollout a policy trajectory and perform back-propagation-through-time [51, 52] with the Adam optimizer [53] to improve the policy parameters. Importantly, the states here are the *commanded* states during demonstration. More details on policy optimization and the exact loss function are included in Appendix D.

# 6 Experiments

**Tasks:** We studied the effect of i) policy structure and ii) the choice of action space by comparing a variety of policy classes across three different tasks (see Fig 1). The first task illustrates prehensile manipulation by requiring the robot to grasp and lift a randomly placed sugar box from the table. The second task illustrates reaching, collision avoidance, and non-prehensile manipulation by requiring the robot to touch a target point on the table while avoiding collision with a randomly placed coffee can and then push the can over to the target. The third task illustrates constrained prehensile manipulation by requiring the robot to precisely maneuver its fingers through a tool tray handle and extract it from a tool box while avoiding undesirable collisions before placing it onto the table.

**Baselines:** We compared the performance of NGF against the following baselines: a fully-connected Neural Network policy (NN), a Riemannian Motion Policy (RMP), a recurrent neural network policy with Long Short-Term Memory units (LSTM), and an End-Effector Neural Network policy coupled with a manually-designed Geometric Fabric for operational space control (EEF-GF). We chose these specific baselines as they represent current practices in imitation learning for manipulation with respect to the choices of policy structure and action space. We provide additional details and justifications in Appendix E, and details about the experiment setup and data collection in Appendix F.

## 6.1 Evaluation Procedure

We trained all policies using the procedure described in Section 5 and deployed them on a physical robot platform to empirically measure their performance in representing the desired behavior implicit in the demonstrations. To be successful, the policies must simultaneously encode approach and grasping behavior by directly coordinating finger, palm, and arm motions for each task.

For each of the first two tasks involving prehensile and non-prehensile manipulation, we trained one instance of each policy class (NN, RMP, LSTM, EEF-GF, and NGF) on 80 demonstrations. We then evaluated each trained policy's ability to generalize to 20 arbitrary object poses on the surface of the table used for demonstrations. Each policy received the object pose estimated by CosyPose [54], a target pose equivalent to the mean pose in the training data, and an initial robot configuration as inputs. For the third task involving tool tray retrieval, we trained one instance of each policy class on 6 demonstrations, in which the toolbox is positioned at 6 different orientations $[10, 30, 50, 70, 90, 110]$ (deg.). We evaluated each trained policy's ability to retrieve the tool tray with the toolbox positioned at 9 different orientations $[-10, 0, 20, 40, 60, 80, 100, 120, 130]$ (deg.). Each policy received the toolbox orientation and the initial robot configuration as inputs.

**Metrics:** We quantified policy performance by evaluating both task *success rate* and the *safe deployment rate*. To compute safe deployment rate and prevent hardware damage, we used each trained policy to generate open-loop trajectories for a horizon of 13 seconds for Task 1, and 30 seconds

for Tasks 2 and 3. We then visualized the robot enacting these trajectories using PyBullet [55] to determine its safety for deployment. We consider a trajectory to be safe if it does not excessively throw the robot into, through, or below the table, and does not lead to violent arm movements. We deployed all safe trajectories on the physical robot using a PD controller. To compute task success rate, we considered an execution to be successful only if the following criteria are met within the deployment horizon. *Task 1*: the robot grasps and lifts the sugar box from the table; *Task 2*: the robot first touches the target point without colliding into the coffee can and then moves the can over to the target location; *Task 3*: the robot grasps and lifts the tool tray out of the toolbox. In addition to these two metrics, we also measured the *imitation error* of each policy (defined in Eq. (12) of Appendix G) to evaluate sample efficiency.

## 6.2 Real-world Task Performance

**Task Success Rate:** The following key observations can be made from the results reported in Fig. 3a: **(i)** Most strikingly, NGF consistently offered the highest level of performance against all baselines and outperformed the best baseline by 40% to 100%, indicating the benefits of our specific architecture and our use of multiple action spaces. **(ii)** EEF-GF had the second highest level of performance by matching other baselines on the coffee can and tool tray tasks and outperforming them on the sugar box task. EEF-GF's good performance is explained by the fact that policies trained with elevated action spaces (in this case, palm pose action) are known to be easier to learn. However, it is important to note that EEF-GF enjoys the benefits of a manually-tuned Geometric Fabric-based operational space controller, and yet preforms worse than NGF which does not require such a controller. This result demonstrates the benefits of learning high-dimensional structured policies directly from data. **(iii)** Both second-order baselines, RMP and NN, surprisingly offer relatively same level of task success rates despite the added taskmap structure of the RMP which aims to introduce beneficial inductive biases. Although less clear, RMP and NN seemed to offer marginal improvements over LSTM, suggesting that second-order policies' ability to encode complex motions is comparable to that of a well-established recurrent structure commonly found in the literature.

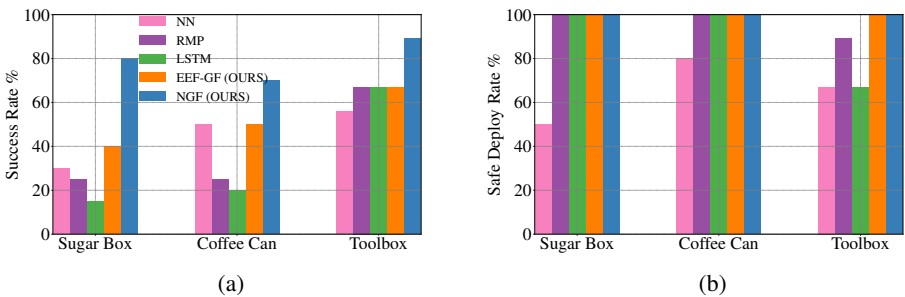

Figure 3: Policy performance on all three tasks based on (a) percentage of deployments that were successful, and (b) percentage of deployments that were safe to deploy on the real robot.

**Safe Deploy Rate:** We report the safe deployment rates for each policy and task in Fig. 3b. Critically, both NGF and EEF-GF were the safest to deploy across all tasks with 100% deploy rates, indicating that Geometric Fabrics-induced structure helps generate smooth and safe motions. Next, RMP had consistently higher deployment rates over NN, indicating that its taskmap structure helps improve safety. Finally, LSTM was comparable to RMP in generating safe trajectories.

## 6.3 Analysis on Sample Efficiency

We also studied the sample efficiency of each policy when learning to perform the first task.

**Imitation Error:** Imitation error captures the difference in trajectories generated by the learned policy and trajectories from the unseen data. We trained each policy on sets of demonstrations with sizes [10, 30, 60], and for every 10 epochs, computed the imitation error on a set of 20 held-out demonstrations. After training for 2000 epochs, we selected the best model based on the imitation error for each policy over 5 random initialization seeds. We report the mean and standard deviation of the imitation error for each policy across 5 random seeds in see Fig. 4a. We note that NGF is the

most sample-efficient as it results in the lowest mean imitation error (and smallest variance) across all quantities of demonstrations.

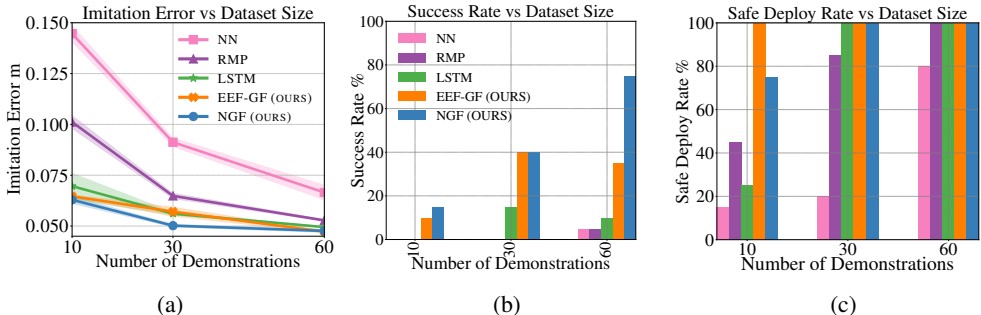

(a)                              (b)                              (c)

Figure 4: Sample efficiency analysis based on (a) imitation error of each policy (solid lines indicate the mean and the shaded area show mean ± standard deviation, over the 5 random seeds). (b) task success rate, and (c) safe deployment rate on physical robot.

**Task Performance and Safety:** Following the procedure in Section 6.1, we report how each policy's task success rate and safe deployment rate change with dataset size in Figs. 4b and 4c, respectively. We note that real-world performance somewhat correlates with imitation error, but not strictly as observed in [2]. Similar to the results in Section 6.2, NGF consistently outperforms all the baselines across all dataset sizes (except for EEF-GF's better safe deployment rate with only 10 demonstrations, perhaps due to its manually-designed controller). Overall, the superior performance of NGF in all three metrics indicate that the additional structure imposed by Geometric Fabrics helps improve sample efficiency in policy learning. We provide an extended discussion in Appendix G.1.

## 7  Final Remarks

**Discussion** The mixture of near-perfect deploy rates and significantly higher task success rates of NGF against all baselines demonstrates the structural power of Geometric Fabrics. We find that EEF-GF is also a very viable architecture which demonstrates how the elevated pose action space when combined with an analytically-derived Geometric Fabric can improve policy learning. Our findings suggest that learning entirely new control structures with multiple action spaces beyond end-effector poses is possible with Geometric Fabrics. Although the policies were deployed open-loop, this is not an intrinsic limitation of NGFs. To illustrate its reactivity, we deployed the NGF policy on the coffee can task in closed-loop. We note that NGF successfully adapted and performed the task, even when we pushed the coffee can during execution (see Appendix H for more details).

**Limitations** Due to the high cost of running real-world experiments, we partially tested at most three random seeds per policy class per task and selected the best seed for full assessment. We acknowledge that experimental and policy variance could produce artifacts in the results. However, the combination of: 1) screening the best policy for each class and task, 2) the low imitation error and associated variance of NGF (see Fig. 4a), and 3) the consistency of NGF excelling across tasks makes it very unlikely that any of the baselines offer better performance in practice. We suspect that the performance of the EEF-GF policy could be improved by including the underlying Geometric Fabric policy inside the training loop. Finally, we deployed the policies open-loop for two primary reasons: 1) object pose detection becomes unreliable during heavy occlusion, and 2) we could assess the safety of the policy's trajectory before deployment, protecting our hardware. We note that training with much larger and diverse datasets will likely enable reliable closed-loop control.

**Future work** Our work opens up a number of avenues for future work, including 1) learning NGF policies from significantly larger and more diverse datasets, 2) learning NGFs that can process raw sensory streams including camera data, and 3) investigating the benefits of NGF for reinforcement learning. We believe that NGF-based learning frameworks have the potential to solve a wide variety of manipulation tasks in closed-loop, enabling us to move away from simplified representations of state (e.g., pose) and towards processing raw proprioceptive, tactile, and visual feedback in a structured manner.

**Acknowledgments**

This research was supported by NVIDIA Research.

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
