# OpenReview forum: "Neural Geometric Fabrics: Efficiently Learning High-Dimensional Policies from Demonstration"
_robot-learning.org/CoRL/2022/Conference — CoRL 2022 Poster_

### Official Review · Reviewer_oe7Z · 2022-06-30

**Originality:** Good
**Technical Quality:** Very Good
**Clarity Of Presentation:** Very Good
**Impact:** 2

**Recommendation:**

Weak Accept: I recommend accepting the paper, but will not argue for my recommendation if the majority of other reviewers have a different opinion.

**Summary:**

The paper proposes a method for learning geometric fabrics from demonstrations for a high DoF hand-arm system. For this purpose the authors define dedicated subspaces suitable for the tasks to be solved and learn the policies as well as the priority metrics in these subspaces manner via behavior cloning.
The authors demonstrate the capability of the proposed method to solve three robotic manipulation tasks, outperforming the chosen baselines in terms of task success rate, safety, imitation error and sample efficiency.

**Issues:**

- I assume the training is done in an end-to-end manner, with the loss in eq. (9) in appendix D defined in the root space. But this is not stated clearly anywhere. I also think it would be beneficial to have more on the training in the main paper and not to defer it completely to the appendix, since it is an integral part for understanding the paper.
- In line 241: I assume the '20 arbitrary object poses' used for evaluation where never seen during training, correct? I am unsure due to the ending of the sentence '... used for demonstrations.', which might refer to the 20 poses or to the table.
- In line 243: Why do you use the mean of the target poses seen in training? Why not use an arbitrary target pose just as it is done for the object poses? Also, do I assume correctly that the target pose is the same for all 20 evaluation runs?
- concerning RMPs: Are the metrics learned? If not, how are they chosen?


**Quality Of The Limitations Section:**

Limitations are addressed clearly

**Reviewer Expertise:**

4: The reviewer is confident but not absolutely certain that the evaluation is correct

**Robotics Focus:**

Sufficient demonstration on hardware

**Strengths And Weaknesses:**

Strength:
- The paper develops a method for learning geometric fabrics from demonstrations and demonstrates the capabilities of the resulting policy.
- The results are compared to those of many baselines in well-chosen experiments and show superior performance in all the defined metrics.
- code is provided in the supplementary material

Weaknesses:
There remain some unclarities, as described in the issues section.



**Summary Of Recommendation:**

I recommend accepting the paper, since it proposes a promising approach for learning geometric fabrics from demonstrations. The capabilities of the method are demonstrated in well-chosen experiments and its superior performance w.r.t. many baselines is shown.

---

> ### Author Response · Authors · 2022-08-19
> **Response to Reviewer oe7Z**
>
> We thank the reviewer for their positive and encouraging feedback. Below we provide answers to specific questions raised under “Issues”.
>
> ### **On end-to-end training**
> Yes, the training was done end-to-end using the indicated loss. We will make this more clear in the updated version of the paper and bring in more training details from the appendix to the main text.
>
> ### **On object poses used for evaluation**
> Thank you for clarifying. Yes, the 20 object poses used for evaluation were unseen during training and were randomly chosen.
>
> ### **On using the mean target pose**
> We indeed use the mean of the target poses in the real-world test. But note that we can use any arbitrary target pose as long as it is not too far away from the training distribution. To ensure consistency, for all 20 evaluation runs, we give the same target pose.
>
> ### **On training the RMP baseline**
> Yes, both the metrics and acceleration policies of the RMP baseline are parameterized by neural networks and are trained end-to-end.

---

### Official Review · Reviewer_1QRC · 2022-07-26

**Originality:** Fair
**Technical Quality:** Fair
**Clarity Of Presentation:** Good
**Impact:** 3

**Recommendation:**

Weak Reject: I recommend rejecting the paper, but will not argue for my recommendation if the majority of other reviewers have a different opinion.

**Summary:**

This paper proposes to learn the parameters of policies encoded via geometric fabrics from demonstrations. Specifically, the paper focuses on learning the policy parameters for a hand-and-arm system, whose motion is generated by two geometric fabrics describing the nominal behavior of the system in task and joint space and a geometric fabric inducing an additional potential function. The difference geometric fabrics are then combined following the principles of RMPflow, where the Riemannian motion policies are replaced by the geometric fabrics. The paper then proposes to encode the geometric fabrics' parameters as neural networks, and to learn them from demonstrations. The paper tests the proposed approach in 3 different manipulation tasks, and assesses its performance with respect to several neural-network-based policies and to a Riemannian motion policy.

**Issues:**

As mentioned in the "strengths and weaknesses" section, the following points should be clarified or improved in the revised manuscript:
- Novelty of the proposed approach;
- Applicability of the proposed approach to different systems;
- Very large number of demonstrations required to learn a task;
- Effective low dimensionality of the main geometric fabrics vs claim of learning high-dimensional policies;
- Clarifications on the content of the main paper;
- References and figures formatting.

**Quality Of The Limitations Section:**

Limitations are not well addressed

**Reviewer Expertise:**

5: The reviewer is absolutely certain that the evaluation is correct and very familiar with the relevant literature

**Robotics Focus:**

Sufficient demonstration on hardware

**Strengths And Weaknesses:**

# Strengths

- The proposed approach is interesting as it allows the learning of the parameters of a geometric-fabric-based policy from demonstrations. This may be beneficial as geometric fabrics may be cumbersome to design by hand.
- The proposed approach is nicely illustrated with 3 real-world experiments involving a complex robotic system (multi-fingered hand mounted on a robotic arm). The paper also made an nice effort in comparing their approach with respect to other state-of-the-art policies on the real system.

# Weaknesses
- The novelty of the paper is rather limited: It mostly follows the ideas presented in [11] (Rana et al, CoRL'20), where the parameters of Riemannian motion policies were encoded as neural networks and learned from demonstrations. As geometric fabrics generalize Riemannian motion policies [12], the contributions of the paper are rather incremental.
- The presented geometric-fabric-based policy is very specific to the system and tasks at hand. Although the presented approach may also be used for geometric-fabric-based policy specified for other systems and tasks, the paper does not mention it. Consequently, it is also unclear how the proposed approach may perform for and scale to different systems/tasks.
- For example, would the proposed approach scale to tasks considering more than 3 geometric fabrics? This is unclear to me, as this would result to an increase of the number of parameters to learn, which may be unfeasible without a very large number of demonstrations (considering the large number of demonstrations used for the tasks considered in the paper, see next comment).
- 80 demonstrations are used for the two first tasks evaluated in the paper. Learning-from-demonstration (LfD) approaches usually use a handful of trials, and aim at facilitating the learning of skills from few demonstrations (including generalizable skills). In that sense, providing 80 demonstrations to essentially learn a picking and a pushing task seem very cumbersome and not adapted for a viable LfD approach. This is even more striking when considering that only 6 demonstrations were provided for the third task considering in the paper.
- Following the previous remark, the analysis on sample efficiency conducted in the first task tend to show that a very large number of demonstrations is needed to obtain a decent performance on the first task. Although the proposed approach is the most sample-efficient according to this analysis, the required number of demonstrations remain considerable. In my opinion, this is a clear limitation, especially when considering that the demonstrations were acquired from a human (via teleoperation).
- The paper claims to learn high-dimensional policies. Although the hand-and-arm system originally has 23 DoFs, the dimensionality of the geometric fabric describing the nominal behavior of the system in task space and of the geometric fabric encoding the potential function is reduced to 8 by considering the position of the end-effector (3 dimensions) and the eigenspace of the hand (5 dimension). These two low-dimensional geometric fabrics clearly encode the most important part of the motion, as the remaining geometric fabric serves as a residual policy, filling in the nullspace of the 8-dimensional space. Therefore, I find the claim of learning high-dimensional policies somewhat exaggerated, as a lower-dimensional space built on prior knowledge is considered for the most important part of the motion.
- Additional details should be given in order to make the paper self-contained. For example, as learning the geometric-fabric-based policies from demonstrations is the main contribution of the paper, the considered loss should appear in the main text. Similarly, it would be beneficial to define the imitation error in the main text. The input of all networks are not detailed in the main paper, while features and states are mentioned as inputs in the Appendix. Additional explanations on this would be appreciated. Which features and states are considered here? Also, the roles of $[M_*, \pi_*]$ (section 4.2) and the decomposition of the terms $\pi_k$ (Figure 2) are not explained.
- The variable $\pi$ is used to describe the geometries (Section 3, 4) and the policy (Section 5), which may be confusing for the reader.
- In Section 2, in the "exploiting structure" paragraph: probabilistic movement primitives ([33], [34]) are not dynamical-system-based approaches. In general, the paper tends to cite long lists of papers without explaining each of them in details (e.g., [1-9], [29-39]). Selectively citing and explaining some of these papers may be more interesting when describing the related work.
- As the paper is concerned with learning from demonstrations and does not consider reinforcement learning, the background section on reinforcement learning does not seem necessary.

## Open questions
- How many parameters are learned for the final policy?
- The low performance of Riemannian motion policies compared to geometric fabrics (and other approaches) in the presented experiments was surprising to me, as geometric fabrics are essentially an improvement on Riemannian motion policies. Is there an explanation for this?

## Cosmetics
- Consider displaying the y-axis from 0 to 100% for all success rate graphs.
- The labels in the figures are generally too small.
- Please improve the format of the references. Many papers cited as arXiv preprints were actually published in conferences and journal. Double check upper cases (e.g., [14] hamiltonian -> Hamiltonian). Some journals are abbreviated, some are not (e.g., [23], [27] vs others). Several citations display twice the year and/or the publisher (e.g., [13], [29], [48]). [10] and [46] cite the same paper. Some papers with many authors are cited usng "et al", some not ([33], [12] vs [7])

**Summary Of Recommendation:**

Overall, the proposed approach is interesting and may be valuable when deploying geometric fabrics on real robots. However, the contributions of the paper are rather incremental and mostly result from a combination of the ideas in [11] and [12]. Moreover, the presented architecture is very specific to the considered hand-and-arm system (especially concerning the design of the underlying geometric fabrics), and it is unclear how it may scale to other systems and tasks. A very large number of demonstrations was required for 2 out of the 3 considered tasks, thus hindering the general applicability of the proposed approach. Finally, several clarifications are required to make the paper self-contained.


--------
**Post-rebuttal comments**

I would like to thank the authors for their detailed answer to my questions and comments. I appreciate the authors' investment in the rebuttal.

However, I think that the main issues of the paper remain after the rebuttal phase. The contributions as initially stated in the paper remain incremental, while I find that the additional contributions stated during the rebuttal are vague and barely reflected in the paper. Moreover, the issues about data efficiency have not been completely answered. Finally, my doubts about the effective dimensionality of the NGF remain as there is still some confusion about the role of the eigenspace for controlling the 16 DoFs of the hand. Although I agree that each eigengrasp controls the 16 DoFs of the hand, the benefit of using an eigenspace is that the final posture of the hand can be expressed as a function of the amplitudes along each of the eigengrasp direction. As 5 eigengrasps are used here, the posture of the hand should therefore be expressed as a function of the 5-dimensional amplitude vector (and not via the 16 original DoFs of the hand), which reduces the dimensionality of the problem.
Therefore, I maintain my initial evaluation.

---

> ### Author Response · Authors · 2022-08-19
> **Response to Reviewer 1QRC: Part 2**
>
> ### **On learning high-dimensional policies**
> We agree that the fabric terms defined in the 8 dimensional space are very important and provide a strong inductive bias that helps when learning from relatively small amounts of data. However, the configuration space fabric terms are also critical and we believe our word choice of calling this configuration space policy a "residual" one underserves its importance. As one can see, the resulting nullspace of the robot given only the fabric terms in this dimensionally reduced space is very large (15 dimensional, in fact), and its importance can be reflected as follows:
>
> > 1. **For arm motion:** the positioning of the elbow can have a significant impact on kinematic reachability, especially close to the body of the arm. However, only 3 dimensions of a 7 DoF arm are actually accounted for by the policy in the 8 dimensional space. This means that the fabric terms living in the 8 dimensional space have no authority over the orientation of the palm or the positioning of the elbow of the arm.
> > 2. **For hand motion:** properly controlling the palm orientation enables accurate fingertip placement and navigation.
> > 3. **For kinematic constraints:** unlike the work in the RMP learning paper, which leverages hand-derived joint limit RMP and obstacle avoidance RMP to dictate the kinematic and environmental constraints, we learn these directly from data through the configuration space fabric.
> > 4. **For theoretic-property:** the configuration space is used to calculate and conserve a Finsler energy, and a damper is learned in configuration space to ensure the convergence.
>
> Overall, the configuration space fabric is critical to the NGF policy and enables it to have full control authority over the entirety of the natural 23 dimensional control space of the robot. Without it, the NGF policy would not be able to solve any of these tasks as, unlike many approaches, it does not rely on a manually-designed operational space controller.
>
> ### **On additional details**
> - **Loss and imitation error:** We will move the loss and imitation error to the main text as we also agree these are core details. We will move the details of the neural network inputs to the main text as well.
> - **Network I/O:** We will move details of the input and output of each network to the main text as suggested. As shown in figure 2 in the paper, there are three types of networks, the first one is a metric net, which takes the state (x, xd) augmented with features (object_pose, target_position) as inputs, the second one is an auxiliary network that are used to construct geometries, which takes the position x augmented with features (object_pose, target_position) as inputs, and the third one is a damping network, which takes the state (x, xd) augmented with features (object_pose, target_position) as inputs.
> - **Figure 2:** We define [M∗,π∗] on line 186, where [M∗,π∗] refers to both the geometric policies and potential policy and their associated priority metric. π_k refers to the learnable geometric policy defined in the k-th subtask space. This is discussed in the caption of Figure 2, but perhaps clarity will be improved if we define these instead in Section 4.2.
> - **Variable π:** We recognize how overloading the variable π may cause confusion. To be clear, π_k is the learnable geometric policy defined in the k-th subtask space, and π_f is the learnable potential policy which exists in some desired task spaces (in our experiments, it only exists in subtask space 1). Their combination along with energization and damping create the complete NGF as discussed in Section 4.3. This complete NGF produces a desired acceleration action, qdd_d (as indicated in Section 4.3) and this desired acceleration action is equal to π_\theta as described in Section 5. We will make this more clear.
> - **Number of learned parameters:** The total number of parameters that are learned is about 88960.
>
> ### **On related work**
> - We thank the reviewer for pointing out that we misidentified ProMPs as a dynamical-system-based approach, and will correct this mistake. We have included an enhanced discussion and comparative results around DMPs which are dynamical-system approaches as discussed above under **Topic 3: Additional baseline: DMP** in the top level comment.
> - We included a discussion of RL methods as they are the most popular approach to learning dexterous manipulation policies. We will compress our discussion on reinforcement learning and expand our discussions on the more relevant references.
>
> ### **On cosmetics**
> Thank you for taking the time to provide such detailed feedback. We will incorporate all suggested improvements.

---

> > ### Comment · Reviewer_1QRC · 2022-08-26
> > **Comments on the authors' response**
> >
> > I would like to thank the authors for their detailed answer to my questions. However, most of my initial concerns about the papers were not answered in the rebuttal.
> > Below are some additional comments.
> >
> > ### Novelty w.r.t. RMPs (topic 1)
> >
> > Thank you for the precisions. However, the answer did not alleviate my doubts concerning the contributions of the paper.
> > Although I agree with the advantages of geometric fabrics mentioned in 1. and 2., I believe that the related contributions may rather be attributed to the original geometric fabrics paper [12].
> > Moreover, the main idea of the paper (making the parameters of GFs learnable from demonstrations via neural networks) is very similar to the idea of [11]. It also seems to me that the procedure to parametrize the GFs essentially consists in replacing their parameters by neural networks, similarly as the parameters of RMPs in [11]. The same network is even used to parametrize the metric (line 188). In other words, the idea an principle underlying the contributions of the papers are not novel, although specific technical changes may have been made to comply with GFs (3., 4.).
> > Although it may be the first time that they are used within GFs (5.), combinations of 3D Euclidean space and eigengrasps are not novel and are rather state-of-the-art to control hand-and-arm systems (see e.g., Ciocarlie et al in R:SS'07, Roa et al ICRA'12, etc).
> >
> >
> > ### On applicability to different tasks and systems
> >
> > Thank you for the clarifications. According to your answer, it still seems that the system was developed specifically for the application and is cumbersome to extend to other systems, although it may be fine for other tasks (you mentioned significant design cycles to actually find adapted task spaces for the specific system and task used in the paper).
> >
> > Concerning the design of NGF with more than 3 fabrics: Given the number of parameters mentioned by the authors, and the high number of data required to train a NGF with only 2 fabrics, it seems to me that training NGF with more fabrics would require a huge number of demonstrations, thus being clearly impractical for robotics applications. Moreover, I think that experiments with more than 2 GFs should be reported in the paper to confirm this answer.
> >
> > ### Data efficiency
> >
> > Although I agree that the number of required demonstrations depends on the complexity of the tasks, it does not seem to me that the complexity of tasks 1-2 compared to task 3 justifies such an increase of demonstrations. It seems rather required by the huge number of parameters that needs to be learned (>88'000), which is clearly a disadvantage of the approach. In this sense, task 3 may require less demonstrations because the networks are completely overfitting the demonstrations. Therefore, the lack of data efficiency should be described rather as a limitation of the paper, especially compared to state-of-the-art LfD approaches such as (non-neural) DMP, ProMP, etc.
> >
> > ### On learning high-dimensional policies
> >
> > In my understanding, the 5-DoFs eigenspace allows the control of all 16 DoFs of the hand. Adding the 3 DoF of the arm controlled in the first task space, it means that 19-DoFs out of 23 are actually controlled by this first task space, i.e., the resulting nullspace is only 4 dimensional (and not 15 dimensional as mentioned in the authors' response). Therefore, it seems to me that using the eigenspace effectively reduces the dimensionality of the configuration space from 23 to 12 (7 DoFs of the arm + 5 DoFs of the eigenspace). Therefore, handling the high-dimensionality of the task space is mostly done by considering the eigenspace, while the NFG effectively handles only 12 dimensions (Or am I missing something here?). Therefore, I still find the claim of learning high-dimensional policies exaggerated.

---

> > > ### Author Response · Authors · 2022-08-27
> > > **Response to Reviewer 1QRC**
> > >
> > > Thank you for the feedback. Here we address these concerns in the same order as they were listed.
> > > #### **On Novelty w.r.t. RMPs (topic 1)**
> > > - The Geometric Fabrics paper is a theory paper that introduced a framework for flexible manual design of robot motions. It does not discuss or investigate whether it could provide a strong inductive bias for learning robot manipulation policies.
> > > - The contribution of this work is not limited to learning NGFs. It includes investigating and leveraging the inductive bias of the Geometric Fabric to learn dexterous manipulation skills with high data efficiency in real-world tasks.
> > >
> > > #### **On applicability to different tasks and systems**
> > > - As presented in the paper, NGFs are designed for the first task, and applied to the second and third task without changing the choice of task spaces. The significant design cycles refer to not only the design of task spaces, but also the entire learning architecture and the specific loss that we used for all policies studied in our work.
> > > - We acknowledge that using more than 2 GFs might require a larger number of demonstrations. However, it is only necessary to use 3 or more GFs if 2 does not solve the task. For more complicated tasks, yes, we might need to use more than 2 GFs to increase the capacity of the policy, which is not different from increasing the neural network size to get more flexibility for all other policies. However, the strong inductive biases leveraged by NGFs will provide better data efficiency as we shown in this work.
> > >
> > >
> > > #### **On Data efficiency**
> > > - There appears to be some confusion due to the fact that our work straddles and contributes to two different communities: i) Researchers utilizing deep IL/RL to learn highly parameterized, high-dimensional policies that solve complex dexterous manipulation tasks with minimal manual design (but at the cost of low sample efficiency and lack of theoretical guarantees), and ii) those who focus on traditional LfD for manipulation to achieve extreme data efficiency with low parameter models (but are either limited to highly constrained problems or depend on expertly designed task spaces and operational space controllers).
> > > - Our work attempts to combine the best of both these areas in order to learn more complex dexterous manipulation behaviors with larger models (compared to traditional LfD for manipulation) and show that models with the inductive biases of Geometric Fabrics offer improved data efficiency and theoretical guarantees (compared to deep IL/RL methods).
> > >
> > > #### **On learning high-dimensional policies**
> > > This is incorrect. Our NGF has a fabric term defined in the full 23 dimensional space so no matter what other terms in reduced spaces we may have, the NGF literally has full control authority over all 23 joints and this behavior is learned.
> > > - The 5 DoF eigenspace has a nullspace of 11 dimensions with respect to the 16 dimensions of the hand, and the 3 DoF end-effector point space has a nullspace of 4 dimensions with respect to the 7 dimensional arm. The combined 8 DoF fabric term can pull on all 23 joints of the robot, but it still has a combined nullspace of 15 dimensions. This means that by itself, it won't be able to encode all the demonstrated behaviors from the data well because it does not have full control authority over all 23 joints.
> > > - The simple example is that a fabric term controlling the 3 DoF end-effector point does not have controllability over the full pose of the end effector even though its existence evokes change in the end-effector pose. This same argument holds true for the fabric term living in the eigenspace of the hand: it can effect change over all 16 joints in the hand, but it does not have full controllability over all 16 joints in the hand. Therefore, we use the fabric term in the full 23 DoF space to create an NGF that has full controllability over all joints of the robot. This 23 dimensional policy resolves this large nullspace in important ways like controlling the orientation of the palm and positioning the elbow.

---

> ### Author Response · Authors · 2022-08-19
> **Response to Reviewer 1QRC: Part 1**
>
> We thank the reviewer for their detailed constructive feedback which was helpful in clarifying and strengthening our arguments.
>
> While we have addressed all major concerns in our top-level comment, below we provide responses to this reviewer’s specific concerns and additional details where necessary.
>
> ### **On novelty**
> We have included a detailed discussion about the novelty of our work with respect to (Rana et al, CoRL'20) under **Topic 1: Novelty w.r.t. RMPs** in the top level comment.
>
> ### **On applicability to different tasks and systems**
> We acknowledge that we didn’t discuss how our approach can be applied to other robot systems and tasks. Thank you for pointing this out. We will include such a discussion in the revised manuscript. For the sake of clarity, we present the key points below:
>
> **Application to other systems:** NGF’s structure is likely relevant to a variety of robot morphologies. For instance, our task space structure can be applied to any serial manipulator with an end-effector. First, we defined a 3D Euclidean space in the end-effector using forward kinematics, and then we augment it with a taskmap defined for the end-effector. For high dimensional end-effectors (e.g. multi-fingered hand), a simple linear taskmap can be obtained by applying PCA to the demonstration data. For low dimensional end-effectors (e.g. 1 DoF gripper), where the linear taskmap map is just a value of 1. In all these cases, the same task map structure can be created as in this work. Fabric terms can be learned in these spaces just as they are in this work. In this sense, one can see how the specific NGF structure uncovered in this paper can apply to a wide variety of robot platforms.
>
> **Application to other tasks:** We would like to emphasize that the same NGF structure was used for three different tasks in this work. In fact, the NGF structure was actually uncovered for Task 1 and directly applied to Tasks 2 and 3, which we did not highlight in the paper. In all cases, the NGF outperformed all baselines. We hope this provides some evidence that this particular NGF structure is relevant across different tasks.
>
> **Different design of NGF with more than 3 fabrics:** We did create a variety of task space structures when designing the NGF and RMP policies and the one reported was the best we found. For example, we created a different taskmap structure for NGFs and RMPs with the following task spaces: 1) the configuration space, 2) three 3D Euclidean spaces on the palm, and 3) four 3D Euclidean spaces, one for each fingertip. This in total had 8 task spaces, each with their own fabric policy. NGF design in this sense is fairly open ended. However, as you pointed out, one can overparameterize an NGF and risk overfitting issues much like any other policy classes, but the structure in NGF provides a strong inductive bias which can help reduce the risk of overfitting issue. We circumvent that here with the two task maps we converged on and the sizes of neural networks making up the fabric terms themselves. If one were to have much larger datasets (in the hundreds or thousands), then one can plausibly learn fabric terms with a large variety of task spaces and achieve good performance with the augmented capacity.
>
> ### **On sample efficiency**
> Please see our detailed discussion on sample efficiency and why different tasks require different numbers of demonstrations under **Topic 4: Data efficiency** in the top level comment.

---

### Official Review · Reviewer_WsPo · 2022-07-31

**Originality:** Poor
**Technical Quality:** Fair
**Clarity Of Presentation:** Good
**Impact:** 3

**Recommendation:**

Weak Reject: I recommend rejecting the paper, but will not argue for my recommendation if the majority of other reviewers have a different opinion.

**Summary:**

This paper proposes to leverage learning-from-demonstration to learn the parameters of a Geometric-Fabric motion policy, parametrized by neural networks. The main motivation of this work is to leverage the structure provided by geometric fabrics in robot motion generation to address the problem of designing dexterous manipulation skills in high-DoF robotic systems. The method is tested on three different manipulation tasks that involving picking, pushing and extraction skills.

**Issues:**

The main issues of this paper are: unclear contributions (it is unclear how much novelty the paper brings when compared to all the RMP and Geometric Fabrics literature), confusing claims on related works and missing references, and the provided experiments. Please see my comments above for suggestions.

**Quality Of The Limitations Section:**

Additional details required

**Reviewer Expertise:**

5: The reviewer is absolutely certain that the evaluation is correct and very familiar with the relevant literature

**Robotics Focus:**

Sufficient demonstration on hardware

**Strengths And Weaknesses:**

**Strengths**
1. Real experiments on hardware:
One of the key aspects of this paper is the real experiments reported in Section 6 and shown in the paper. On the one hand, it is very clear how introducing the structure of geometric fabrics brings benefits when learning dexterous manipulation skills, which is very clearly observed in Fig. 3. On the other hand, the information provided in Appendix and the available code may provide the opportunity to reproduce the results reported in the paper and/or employ the proposed method in other works.

2. Facilitate the design of geometric fabrics via LfD:
It is definitely important to highlight that the design of the parameters of geometric-fabric policies may be tedious, and leveraging LfD to address this problem brings some benefits when using this robot motion generator in practical settings.


**Weaknesses**
1. Unclear contributions:
   * The paper clearly states three contributions in lines 59-65, claiming novelty on the use of LfD for geometric-fabrics policies, thorough analysis of the policy structure, and the first framework to train and generate joint-space motions on high-DoF dexterous manipulation using learning methods. Below I explain the reasons why I disagree with some of these claims:
      - Using LfD and Geometric fabrics seems to be relatively new. However, the combination of LfD to learn RMP parameters (a very similar approach, as acknowledged in the paper) was already proposed by Rana et al [11]. There is nothing particularly new in the design of the networks or learning process that is specially designed for Geometric Fabrics. In this sense, it is hard to see this as a novelty. Going beyond this, Geometric fabrics is an approach proposed in [12] and [13], the tree-like structure to combine different policies was proposed in the RMPflow work [45] and also improved in [10].

     - Analysis of the importance of the policy structure is somehow confusing. On the one hand, I agree that comparing agains FF and LSTM networks is a good way to show that policies with some meaningful structure provide benefits when it comes to learn motion policies. However, the comparison against RMP, which is also an approach that provides a strong policy structure is not analyzed in detail. It is unclear why such big differences exist given the fact that RMPs may also be learned from demonstrations and have shown successful experiments in past works. The paper should provide clearer evidence or a deep discussion on the possible reasons of the observed performance differences.

2. Related work:

   Although the paper does a good job by reviewing different works using IL and RL to learn dexterous manipulation skills, there are some points I think deserve special attention:

   - Two recent works on learning dexterous manipulation (on high-DoF robots) from demonstration are missing [a,b] (which I think are very close, in practical terms, to the concepts and experiments reported in this paper):
      [a] A. Gupta, C. Eppner, S. Levine and P. Abbeel, "Learning dexterous manipulation for a soft robotic hand from human demonstrations," 2016 IEEE/RSJ International Conference on Intelligent Robots and Systems (IROS), 2016
      [b] P. Ruppel and J. Zhang, "Learning Object Manipulation with Dexterous Hand-Arm Systems from Human Demonstration," 2020 IEEE/RSJ International Conference on Intelligent Robots and Systems (IROS), 2020.

   - In the review of IL approach, the paper claims that one of the weaknesses of the related works is the data efficiency, yet the proposed approach in this paper was tested with 80 demonstrations for one of the experiments, which does not seem any different from other IL approaches.

   - In lines 102-104, the paper claims that NFG provides a more “appropriate inductive bias” than RMPs. But it is unclear what the paper means with “more appropriate”. This claim needs to be clarified.

   - In more general terms, the paragraph on exploiting structures does not provide any insights on the differences between the structure that NFGs bring when compare to more classical motion primitives such as DMP and ProMPs, which are cited but not discussed.

3. Experiments:
As pointed out above, the comparison against FF and LSTM networks clearly show the advantages of having structure in the robot policy representation, this is a great result to report. However, I would like to suggest two important improvements in the experiments:
   - Figure 3 and 4 are all about showing the benefits of including structure for task performance and data efficiency. However, if there is an approach in the literature that stands out as a method that is data efficient and includes structure is DMPs, cited in this paper. DMPs provide a second-order DS structure, can be learned from a single demonstration (in some mild cases) and are mainly learned using LfD. So, this seems to be a very good baseline to use to compare against to. Either if the DMP is learned in task or joint space, this may provide some additional insights on the performance of NFG against other approaches that provide strong structure.
    - It is unclear if RMPs parameters are learned as proposed by Rana et al, or if this paper considers manually-designed RMPs for comparison purposes.
    - Figure 4, in my opinion, shows that NFG are **not** data efficient. They seem to be better than FF and LSTM, but considering success, in terms data efficiency, when using 60 human demonstrations is not data efficient in LfD literature (consider DMP or ProMPs, which can be trained with a handful of demonstrations). More importantly, real applications of LfD approaches that demand the human teacher to provide more than 10-15 demonstrations are not practical. This should be considered a limitation of the approach and discussed in the corresponding section.

*Lack of details in proposed approach*
- Clarify why the metric-weighted projection matrix $\mathbf{P}_e$ boils down to an identity-matrix metric in line 193.
- Line 217 refers to the appendix to understand the policy optimization and loss function, but in my opinion, both are core components of the proposed approach and should appear in the main text.



**Summary Of Recommendation:**

The paper leverages LfD to learn the parameters of Geometric-Fabric policies. In this regard, the paper does not clearly state the main differences w.r.t all the previous works on RMPs and Geometric-Fabrics, as some of them (in particular Rana’s work) highly overlap on the ideas and concepts that this paper builds on. Along a similar line, due to the high overlap with previous works, it seems that several claims about novelty are hard to see. Last but not least, the experiments fall short on providing strong evidence on data efficiency, and there is a lack of comparison against more classical movement primitives, that theoretically speaking, may also provide strong structure and data efficiency, as claimed for NFGs.

------

**Post-rebuttal recommendation**

Thank you very much for the rebuttal and interesting discussion. I appreciate very much the time and effort invested during the rebuttal phase. Below you can find my thoughts on my final recommendation:

1. As I stated in the rebuttal, GFs and RMPs are very similar geometric control approaches, with the former being a sort of generalization of the latter. As such, nearly every extension made about RMPs (RMPflow, RMP2, RMP learning via demonstrations) can be applied to GFs. In fact, the paper leverages RMPflow computations and similarly, the proposed extension to learn the GF parameters from demonstrations is very close to the RMP extension of Rana et al.,(2019). In this regard, I find the authors' reply unsatisfying.

2. Another aspect that is not fully addressed is sample efficiency. In my experience, well-known LfD approaches are easily trained with 10-15 demonstrations (e.g., DMPs, GMMs, CNPs, ProMPs, to mention a few) for fairly complicated tasks. Some of these approaches, extended to complex settings like Riemannian manifolds, have been trained with such small datasets. So, in this regard, 60+ demonstrations do not align with sample efficiency claims. Thus, the authors' rebuttal was unsatisfying.

3. I fully agree with the authors' on that GFs can be seen as inductive bias in robot motion skill learning (as discussed in my rebuttal). However, this is not specific to this paper, but it is a contribution of the GFs papers themselves. On a similar line, if such inductive bias were a game changer when learning this kind of policies, the paper falls short on providing evidence on this regard: RMPs, which also bring inductive bias to the learning procedure, seem to perform similarly w.r.t NN or LSTM (in some cases). Then it is unclear why the inductive bias that RMPs bring to the problem do not provide improvements over these baselines and why GFs do instead.

4. Last but not least. I think the work reported in this paper is very interesting, it has good merits, as I pointed out in my review. But I would like to also second the feedback of reviewer 1QRC, which highlights other aspects that I overlooked in the paper and have not been properly addressed.

Then, I keep my recommendation.

---

> ### Author Response · Authors · 2022-08-19
> **Response to Reviewer WsPo**
>
> We thank the reviewer for their detailed constructive feedback which was helpful in clarifying and strengthening our arguments.
>
> While we have addressed all major concerns in the top-level comment, below we provide responses to this reviewer’s specific concerns and additional details where necessary.
>
> ### **On clarity of contributions**
> Thank you for pointing out that our contributions could be better articulated. Please see our detailed discussions on i) the novelty of our work over RMPs under **Topic 1: Novelty w.r.t. RMPs**, and ii) why RMP’s performance seems to be poor in our tasks under **Topic 2: RMP’s poor performance** in the top-level comment.
>
> ### **On improved sample efficiency**
> It’s true that many IL approaches have been able to learn manipulation skills from a handful of demonstrations. But it is important to note that these approaches tend to consider end-effector skills on serial link manipulators, which are much simpler than the dexterous manipulation skills demonstrated in our paper. The number of required demonstrations required for good performance will indeed depend on a variety of factors including: 1) policy structure, 2) task complexity, 3) robot complexity, and 4) reliance on expertly-designed operational space controllers. We isolate the effect of policy structure on task performance by fixing the last two. We see that across all our tasks (with varying complexity) that NGFs outperform all baselines, lending credence to NGFs improving data efficiency. Moreover, we do actually include a task (the tray extraction task) that only requires 6 demonstrations to achieve good performance. This is in line with existing works reporting only a handful of demonstrations are needed for good performance. However, this result is achievable only when the task is much simpler. In contrast, the first two tasks scatter the object randomly across the table and require much more precision in the coordination and placement of the robot’s fingertips. The first task even requires appropriate grip coordination. Further, note that NGFs do not rely on bespoke operational space controllers unlike a vast majority of existing approaches in IL-based manipulation. Please see **Topics 2, 3, and 4** in the top-level comment for a more detailed discussion.
>
> We also thank the reviewer for bringing the missing references to our attention and will include them and discussion thereof in the next iteration of the paper.
>
> ### **On better inductive bias**
> We agree that our initial phrasing “more appropriate inductive bias” is vague and confusing. We will change the wording to "additional structural inductive bias" and explain what specific structure is introduced and how it helps.
>
> Specifically, NGFs require the following four components: 1) a system geometry, 2) a Finsler energy, 3) a potential function, and 4) a strictly positive damper in the root space. This specific machinery promotes path consistency of the NGF, guarantees stability of the motion, and encodes much of the rich behavior into the geometry, circumventing issues around conflicting potentials. Reference [12] demonstrates how this machinery enables hand-engineered Geometric Fabric policies to outperform RMPs. Ultimately, Geometry Fabrics are more constrained than RMPs, but these constraints can provide useful inductive biases as we show empirically in the experiments in this paper.
>
> ### **On comparison with DMP-based methods**
> Thank you for suggesting that comparisons with DMP will help communicate the benefits of our approach over other structured approaches.
>
> We agree that we need more discussion on DMP as it is a second order policy and a natural baseline. In the interest of space, we exclude comparisons with ProMPs as they are not dynamical systems. The main differences between DMPs and NGFs are that DMPs are linear dynamical systems with nonlinear perturbations that phase out over time. In contrast, NGFs are stable, autonomous, nonlinear second order differential equations with an emphasis on path consistency. In fact, as discussed in [12], NGFs generalize all classical mechanical systems which are used, for instance, to model the equations of motion of physical robots. We will augment discussions like these in the next iteration of the paper.
>
> We have also included results on NDP (a recent method based on DMP suggested by the Area Chair). Please see discussion under **Topic 3: Additional baseline: DMP**.
>
> ### **On training RMP parameters**
> We do not use any manually designed RMPs in this work. We learn the RMP parameters end-to-end from data. Please see **Topic 2: RMP’s poor performance**  for a thorough discussion of the differences between our work and that of Rana’s.
>
> ### **On metric weighted projected matrix**
> This analysis is covered in Theorem IV.5 of [12]. We referred to equation (10) of [12] in our paper on line 194 to refer the reader to more details on this analysis. If preferred, we can bring these details to our Appendix from [12].

---

> > ### Comment · Reviewer_WsPo · 2022-08-26
> > **Response to authors' reply**
> >
> > Dear authors,
> >
> > Thank you very much for your very detailed answer and the effort on making your best to clarify and address some of my comments/concerns. I really appreciate the time invested by the authors on this rebuttal.
> >
> > After having read the paper and the authors' reply to my review and other reviewers' feedback, I find it difficult to have a concise picture and understanding of the specific contributions of this paper. The reply sometimes implied contributions at experimental levels, sometimes at methodical levels (e.g., inductive bias from GFs), and sometimes about system integration aspects. Please do not get me wrong, I do think that a scientific paper may cover all these fronts, but my general opinion about the reply is that the contributions are not conveyed clear.
> > Last but not least, it is unclear for me how all the answers provided in the reply will be incorporated in a revised version of the paper, which makes it difficult to assess how much the paper could improve.
> >
> > I elaborate my main concerns below:
> >
> > 1. I agree that Geometric Fabrics should not be seen as a competitor of RMPs, but they do share several technicalities, as acknowledged in the original Geometric Fabrics papers, that make them conceptually very alike. In this regard, some of the open parameters to choose/tune are similar to those that one may need to find for RMPs. Therefore, the idea of learning such set of parameters using LfD is not novel per se. Moreover, the improvements of using NGFs when compared to RMPs, in the reported experiments, show that the inductive bias of the former may be better for the systems and tasks considered in the paper, but this is a contribution of the original papers that proposed Geometric Fabrics, not a contribution of this paper (when one considers specifically the benefits of inductive bias for the compared models).
> >
> > 2. The authors' reply claims that this work *"represents the first successful attempt at learning generalizable dexterous manipulation policies in the configuration space directly on a high-dimensional physical robotic platform purely from a limited number of demonstrations"*. On the one hand, this seems more like a description of the paper rather than the technical/experimental/theoretical contributions of the paper. It is well-known that imitation learning for high-dimensional robotic systems has been addressed by several researchers in the past [1,2] (to give some examples), with successful results, and working on configuration spaces of similar dimensionality as the one addressed in this paper. Obviously, the use of NGFs may be novel in this regard, but the claim given by the authors seems to ignore quite few works on imitation learning that addressed high-dimensional problems in the past. In this regard, my recommendation is to rephrase this contribution of the paper, to focus more on the steps/benefits/advancements of learning GFs using LfD.
> >
> >    [1] Grimes, D.B., Rao, R.P.N. (2009). Learning Actions through Imitation and Exploration: Towards Humanoid Robots That Learn from Humans.
> >
> >    [2] Schaal, S., Atkeson, C.G. & Vijayakumar, S. Scalable Techniques from Nonparametric Statistics for Real Time Robot Learning. Applied Intelligence 17, 49–60 (2002).
> >
> > 3. I appreciate very much the new baseline added in the paper, although using NDPs does not correspond to benchmark against DMPs, as the learning methods are very different. So, my comment about comparison against a very basic baseline (that is indeed very sample efficient for complicated motions) is still unaddressed.

---

> > > ### Author Response · Authors · 2022-08-27
> > > **Response to Reviewer WsPo**
> > >
> > > Thank you for the feedback. Here we address these concerns in the same order as they were listed.
> > > #### **On contribution**
> > > 1. The Geometric Fabrics paper is a theory paper that introduced a framework for flexible manual design of robot motions. It does not discuss or investigate whether it could provide a strong inductive bias for learning robot manipulation policies.
> > > 2. The contribution of this work is not limited to learning NGFs. It includes investigating and leveraging the inductive bias of the Geometric Fabric to learn dexterous manipulation skills with high data efficiency in real-world tasks.
> > >
> > > #### **On previous work**
> > > We thank the reviewer for pointing out these references, and we agree that previous work has shown cool results in learning skills for robots from demonstrations. However, they were mostly restricted in what they learn because they address a specific subproblem (single modal reactive skill learning). None of the previous learning work pointed out here is really applicable to dexterous manipulation tasks because
> > >   - They are learning simple single mode differential equations. E.g. it would amount to a single way of picking up an object (single grasp), and any ability to switch grasps based on object pose would have to be hard coded.
> > >   - They are addressing problems that are either taking high dimensional inputs (e.g. 7D positions + 7D velocities + 7D accelerations) but only output commands for a single joint or learning for simple end-effector behaviors with the extended Jacobian method.
> > >
> > > General imitation learning from human data requires flexible representations to handle the multi-modality of behaviors natively expressed in the demonstrations. However, Standard techniques (NN / LSTM policies and even RMP policies using straightforward imitation) in that context don't have practical data efficiencies for training on physical hardware. We empirically show that our method NGF, by leveraging geometric fabrics, does.
> > >
> > > #### **On DMP**
> > > NDPs are DMPs with components parameterized by neural networks. This paper studies policy classes parameterized by neural networks in which case NDPs represent the DMP policy class. Moreover, we intentionally lock the training mechanism across all policy classes so that we can isolate the effect of policy class towards performance. Therefore, we train the NDP in the same way as all other policies in this paper. Finally, it is not clear what training mechanism you have in mind for DMPs. DMPs are a specific policy structure and it can be trained using a variety of training mechanisms including reinforcement learning. This paper fixes the learning mechanism and studies performance across a variety of policy classes.

---

### Official Review · Reviewer_zkro · 2022-08-01

**Originality:** Very Good
**Technical Quality:** Very Good
**Clarity Of Presentation:** Very Good
**Impact:** 4

**Recommendation:**

Weak Accept: I recommend accepting the paper, but will not argue for my recommendation if the majority of other reviewers have a different opinion.

**Summary:**

The authors propose a new learning from demonstration architecture that creates a structured policy capable of controlling robots with high DoF from a limited number of demonstrations. The paper presents the architecture and learnable parameters of Neural Geometric Fabrics (NGF) by first introducing the relevant prior work on Geometric Fabrics. An extensive comparison of NGF and several baselines is presented demonstrating the success rate and safe deployment of NGF on three tasks.

**Issues:**

* In Figure 2, the numbering in the caption does not easily follow the graphic. For readability could numbering or steps be added to the graphic.

**Quality Of The Limitations Section:**

Limitations are addressed clearly

**Reviewer Expertise:**

3: The reviewer is fairly confident that the evaluation is correct

**Robotics Focus:**

Sufficient demonstration on hardware

**Strengths And Weaknesses:**

# Strengths
* The paper is well written and shows a clear progression from the relevant prior work to the proposed architecture. The authors make a good effort to simplify the complexity of the method for the reader.
* The authors select 4 baselines for comparison with NGF and provide a comprehensive evaluation of the results.
* The supplementary material includes a video verifying the robot deployments and further evaluations of the baselines and NGF method along with architecture parameters and algorithms.

# Weaknesses
There are no large or obvious weaknesses to this paper. However, another baseline for comparison that induces structure for learning motion trajectories would be helpful for the reader to validate the improvements NGFs make to the learning from demonstration area. The only method that comes to mind is "Conditional Neural Motion Primitives" although there are likely better ones.

**Summary Of Recommendation:**

This paper introduces a novel, structured method for learning from demonstration and proves its sample efficiency and ability to control robots with high degrees of freedom. The inclusion of multiple baselines and a thorough experimental section make this a considerable contribution.

---

> ### Author Response · Authors · 2022-08-19
> **Response to Reviewer zkro**
>
> We thank the reviewer for their positive and encouraging feedback.
>
> We agree that adding another baseline would strengthen the paper. As recommended, we now have included NDP (a recent method based on DMPs suggested by the Area Chair) as an additional baseline following the literature. Please see response to **Topic 3: additional baseline: DMP** in the top level comment.
>
> Further, we will make the suggested changes to Figure 2.

---

### Author Response · Authors · 2022-08-19
**Topic 1: Novelty w.r.t. RMPs (Reviewers WsPo and 1QRC)**

**Key similarities with RMPs**
> 1. Both RMPs and NGFs (our method) are dynamical-systems-based structured policies which can be learned from demonstrations.
> 2. NGFs leverage a tree structure of subtask spaces like RMPs.

**Key differences between NGFs and RMPS**

It is important to note that RMP does not necessarily represent a competing approach to our NGF. In fact, Geometric Fabrics can be viewed as specialized versions of RMPs with more effective structural inductive biases. Indeed, our key contributions are to enable learning Geometric fabrics directly from demonstrations and to demonstrate that the inductive bias in our approach is considerably more data efficient than that introduced by RMP within the context of high-dimensional dexterous manipulation skills. To this end, we solve a number of challenges and our approach results in the following benefits over RMPs:
> 1. **Geometric fabrics provide a unique and beneficial inductive bias.** The combined works of [12] and [13] set the theoretical foundations culminating in second order systems that can be classified as a Geometric Fabric. A Geometric Fabric requires four main components: 1) a system geometry, 2) a Finsler energy that we will force the geometry to conserve, 3) a potential function, and 4) a strictly positive scalar acceleration damper. This combined machinery was only recently introduced in [12] and [13] and no prior work has demonstrated that this structured class of policies can be learned from demonstrations.  Our experiments show that the coordinated high-dof dexterous manipulation demonstrations are well-modeled as geometries, which separate the motion (paths) from speed of traversal (an inductive bias that allows the policy to learn similar motions from trajectories that may have the same shape but different speed profiles).
> 2. **RMPs were not developed for high-dimensional dexterous manipulation policies.** The original paper demonstrates the effectiveness of RMP on 3D end-effector skills for a fixed task on a 7 dof manipulator, and its learned networks do not consider task features such as the object pose. In stark contrast, NGFs can effectively learn complex dexterous manipulation skills on a 23-dof hand-arm system while accounting for task features.
> 3. **Leveraging the theory of geometric fabrics in a learning context requires nontrivial design not addressed in [12] and [13].** We introduce how to parameterize the NGF with neural networks while ensuring that the theoretical guarantees of Geometric Fabrics are maintained. Specifically, we construct two HD2 geometric policies, one potential policy, their associated priority metrics, and a damper via neural networks. We also choose to energize these geometries via a specific energy established in configuration space and establish a novel tree of spaces that these components reside within (more discussion below).
> 4. **Our training technique and design of loss functions represent steps beyond prior work.** These decisions were not investigated or considered in existing RMP training papers, and required significant iteration before we converged on the proposed learning algorithm. These choices greatly impacted performance.
> 5. **Our choice of action spaces (the configuration space and concatenated PCA space for the hand and 3D Euclidean space at the palm point) is nontrivial and is unique to our work.** We spent significant design cycles iterating on which task spaces performed best for both RMP and NGF policies. We believe that these insights will be highly relevant to future efforts aimed at learning dexterous manipulation policies, even if a completely different policy parametrization is utilized.
> 6. **The novelty of our work (or that of RMP) does not reside on task space trees.** In fact, trees of task spaces have a long history in the robotics literature dating from much earlier than RMPs or geometric fabrics. RMPs and especially RMPflow introduce a data structure that’s helpful for leveraging pullback in solving structure task space least squares problems, and that’s used in geometric fabrics and our work here (geometric fabrics actually add non-trivial machinery based on the theory in [12] and [13]). But the use of transform trees here isn’t our focus. Our work demonstrates that the inductive bias of geometric fabrics is pivotal to achieving data efficiencies low enough to make learning high-dof dexterous manipulation from human demonstrations on physical hardware possible.

For these reasons, we believe that our work contributes both practical insights and a novel methodology for learning highly-structured dexterous manipulation policies from demonstrations. In this paper, we show that the structure induced by NGFs significantly outperform all baselines, including RMPs. Our work represents the initial steps in investigating the practical benefits of strong, theoretically-derived structure on learning systems.

---

### Author Response · Authors · 2022-08-19
**Topic 2: RMP’s poor performance (Reviewers: WsPo, 1QRC)**

> 1. **The RMP has been shown to be successful only on considerably simpler tasks, which are effectively goal-directed reaching motions without prehensile manipulation.** The objects for which the robot reaches have a single degree of freedom (ours have up to 3 degrees of freedom) and the test distribution over this single degree of freedom is much smaller than ours. For instance, we place the objects almost anywhere on a table in front of the robot for two tasks and rotate them within a 360 degree interval.
> 2. **The robot used in the RMP learning paper has only 7 degrees of freedom, while our robot has 23.** This difference in size of action space makes learning considerably more difficult.
> 3. **RMP-based policies rely on considerable manual engineering efforts** including: 1) manually designed damper, 2) potential function is partially hand crafted and the remaining portion is parameterized as a function of demonstration data. It’s not clear how to extend this to our case of demonstrations with very wide distributions (a limitation pointed out in the existing RMP learning paper), and 3) usage of hand derived joint limiting RMPs and obstacle avoidance RMPs. This significant manual construction and specific structure of the potential function, the reduced task complexity, and reduced robot complexity enabled training the learnable portions of the complete RMP policy to solve the specified tasks.
> 4. **It is not trivial to directly apply RMPs to our problem of interest: dexterous manipulation.** We acknowledge that RMPs are a very broad class of second order dynamical systems with metrics that are general positive definite functions of position and velocity and acceleration policies are general functions of position and velocity. Consistent with the original paper [41], our RMP baseline parametrizing these metrics and acceleration policies via neural networks as functions of position, velocity, object pose, and target position. To ensure fairness to RMPs, we do not hand engineer damping, potential functions, or other RMP terms, and instead learn our RMP baseline end-to-end entirely from data just like all our other policy classes.

In summary, due to the lack of additional manual engineering efforts, increased complexity of our tasks, and increased complexity of our robot system, our RMP policy does not perform as well as it tends to perform when handling considerably simpler tasks.

---

### Author Response · Authors · 2022-08-19
**Topic 3: Additional baseline: DMP (Reviewers: zkro, WsPo, 1QRC)**

We agree that adding a DMP-like baseline could further strengthen our experiments. We have now trained an NDP (a recent method based on DMP suggested by the Area Chair) following the formulation in the suggested paper with our loss function.
> 1. **How NDP works:** When used in imitation learning, NDP only evaluates its neural network components once at the beginning given object pose and target position to produce the g, w, alpha, and beta terms in the DMP. These terms are held fixed and the DMP is forward integrated to produce the command trajectory. This strategy is the same as used in the NDP paper for the imitation learning setting.
> 2. **Pros and cons:** This structure of NDP is good for avoiding distribution shift issues that other methods suffer from and alleviating the overfitting issue when there are only a few demonstrations. However, it diminishes its capacity in encoding the fine details of the behavior, which are very critical in dexterous manipulation with high-dof systems.
> 3. **Experiment results:**  As shown in the imitation error plot (**see Figure 1 in the attached pdf file**), thanks to the strong inductive bias, NDP outperforms NN when trained with less than 30 demonstrations. However, due to the lack of expressivity, NDP is outperformed by NN when trained with 60 demonstrations. Overall, the NDP has the worst imitation error of all policies in the higher data regime and significantly worse imitation error than the LSTM, EEF-GF, and NGF policies in general. Given these results, we expect the NDP to have real world performance somewhere between the NN and RMP policies, neither of which can compete with the NGF policy in the real world experiments.
> 4. **Ablation study:** We also ran ablation studies across different neural network sizes just like all our policies (**see Figure 2 in the attached pdf file**) and decided on the median network size [256, 128] with 10 RBFs given its best performance in imitation error.

---

### Author Response · Authors · 2022-08-19
**Topic 4: Data efficiency (Reviewers: WsPo, 1QRC)**

With regards to data efficiency, we would like to point out that the number of training demonstrations required for good performance on any particular task naturally correlates with the **complexity of the task**. Existing methods that can afford to learn dynamical systems that recreate the expert data with just a handful of demonstrations are leveraging **lower dimensional systems, tighter distributions of expert trajectories, and tasks that require less precision**. For instance, our third task of extracting a tray from a toolbox is relatively simple compared to the other tasks, as the toolbox is only rotated along one axis, and the robot starts from the same initial configuration. As a result, the distribution of expert trajectories is much tighter and we can achieve good performance with just 6 expert demonstrations. For any tasks that are similar in nature, we also expect to only require a few demonstrations to achieve good performance. However, for more complicated tasks like grasping and lifting a low profile box from nearly anywhere on a table (our first task) or pushing a coffee can to a target point from nearly anywhere on a table (our second task) requires more demonstrations to effectively learn the requisite intricate movements. Further, the trajectories generated by the expert on these kinds of tasks can be highly variable and this makes learning the true expert behavior more difficult.

To our best knowledge, **no work in the past has shown success in learning dexterous manipulation skills for high-dimensional robotic systems (20+ DoF) purely from demonstrations without relying on a bespoke operational space controller.** In particular, for a robotic arm and multi-fingered hand system, the success of learning high-dimensional dexterous manipulation skills usually requires the use of a hand-derived operational space controller (OSC), as it can help avoid the difficulty in learning the arm motion. However, such a OSC is not required in our framework, as we can learn both the arm and the hand motion simultaneously from demonstrations. Moreover, our EEF-GF policy is related to other works that learn policies on top of OSC controllers and even in this case, the EEF-GF policy could not beat the NGF trained end-to-end.

We again thank the reviewers for their feedback and systematically addressed their remaining concerns in our following replies.

---

### Author Response · Authors · 2022-08-19
**High-level Comment**

We thank the reviewers and the meta reviewer for their detailed and constructive feedback and take this opportunity to clarify some points made with respect to the paper. We have organized our response as follows:
- This top-level comment which contains an executive summary and discussion around four specific topics mentioned in the meta review.
- Individual responses to each reviewer’s comments to address any outstanding concerns.

### **Executive Summary**
Our work is focused on imitation learning of complex, contact-rich manipulation policies in C-space, such as the 23 DoF space of a robotic arm augmented with a dextrous hand (used in our experiments). **To the best of our knowledge, our work represents the first successful attempt at learning generalizable dexterous manipulation policies in the configuration space directly on a high-dimensional physical robotic platform purely from a limited number of demonstrations.** Existing methods either i) limit their focus to considerably simpler skills and assume a well-designed operational space controller in order to provide theoretical guarantees and sample efficiency (LfD methods, including RMP, that learn end-effector skills on serial-link manipulators, such as goal-directed and point-to-point motions) or ii) assume access to significant amount of data or are limited to simulation in order to handle complex skills and high-dimensional robots (Deep RL methods that learn dexterous manipulation skills for multi-fingered hands). Our work contributes to and improves upon both these categories of existing approaches by developing a sample-efficient approach to learning dexterous manipulation skills directly on a high-dimensional physical hand-arm system with strong theoretical guarantees and generalization capabilities.

We investigate how data efficiency varies across different structured policies defined directly in the C-space (building off the RMP2 framework which the authors of the RMP learning paper developed to generalize their own work) with the goal of finding a structure that’s flexible enough to express proficient high-dimensional policies while also encoding sufficient inductive bias to achieve practical data efficiencies for imitation on physical hardware. Our experimental analysis shows that the recent framework of Geometric Fabrics hits that mark.

---

### Meta-Review · Area_Chair_97oj · 2022-08-15

**Recommendation:** Accept (Poster)
**Confidence:** 3

**Metareview:**

Reviewers agree that this is an interesting paper, however, they also identified a few issues with the paper:
- The relation and novelty w.r.t to the RMP paper seems unclear as both approaches seem to be very similar.
- The comparison to the RMP approach is missing many details and one reviewer was questioning the poor performance of RMP as they performed well in other papers
- Additional baseline should be considered such as DMPs. DMPs have also been extended to neural network architectures, see [1].
- The data efficiency of the method has to be discussed and evaluated in more detail

These issues (as well as other reviewer comments) should be addressed carefully in the rebuttal.

[1] https://papers.nips.cc/paper/2020/hash/354ac345fd8c6d7ef634d9a8e3d47b83-Abstract.html

Rebuttal Update: Unfortunately, the reviewers could not come to the same opinion on this paper even after a longer discussion period. 2 reviewers do see the following issues:
- in terms of novelty of the approach as the given architecture is a special case of an RMP and RMPs have already been "neuronalized". Further, they argue that the seen advantages of the architecture is a contribution of the original GF paper (which has no neural learning architecture in it). Even after a longer discussion with both reviewers, I can not follow their argument as the given paper is the first to put GFs in a neural architecture and show that it is more effective than a standard RMP architecture. I do see that as a significant contribution.
- Furthermore the reviewers raised the issue that the approach is using eigengrasps to reduce the dimensionality of the problem. Even with eigengrasps the dimensionality of the problem is high higher then used in many other papers, so I also can not follow this argument.
- Another issue was raised in terms of data complexity as the approach needed 80 demonstrations which is, in comparison to other methods such as DMPs a quite high number. However, plain DMPs are a much simpler approach that would never scale to this complexity. The authors compared to Neural architectures of DMPs in the rebuttal, so I think this point is also addressed.

I agree that the contribution of the paper (differences to a standard RMP) should be made clearer in the paper and the different parts of the architecture should be ablated to see why we get such a performance difference (as part of future work). However, given that the paper offers an interesting contribution for data-driven motion generation and offers real-robot experiments of high complexity, I would actually side with the two positive reviews and would like to see the paper published.





**Best Paper Nomination:**

No

---

> ### Author Response · Authors · 2022-08-19
> **Response to Area Chair**
>
> We thank the reviewers and the Area Chair for their detailed and constructive feedback. We have addressed all major concerns in the top-level comment, please see **Topics 1, 2, 3, and 4** for a detailed discussion around four specific issues mentioned in the meta review.

---

> ### Author Response · Authors · 2022-08-27
> **Response to Area Chair: clarification on fundamental misunderstandings**
>
> We thank the reviewers for their responses. However, we respectfully disagree with their key arguments. We believe that there may be some fundamental misunderstandings about the paper which we clarify below:
> 1. Reviewers state that learning geometric fabrics is not novel and reported improved performance is a novelty of the original Geometric Fabrics paper and not this one.
>     - **The Geometric Fabrics paper is a theory paper that introduced a framework for flexible manual design of robot motions. It does not discuss or investigate whether it could provide a strong inductive bias for learning robot manipulation policies.**
>     - **The contribution of this work is not limited to learning NGFs. It includes investigating and leveraging the inductive bias of the Geometric Fabric to learn dexterous manipulation skills with high data efficiency (NGFs outperform all baselines) without the need for an expertly designed operational space controller (as required by the references cited by the reviewers).**
>
> 2. There appears to be some confusion due to the fact that our work straddles and contributes to two different communities: i) Researchers utilizing deep learning to learn highly parameterized, high-dimensional policies that solve complex dexterous manipulation tasks with minimal manual design (but at the cost of low sample efficiency and lack of theoretical guarantees), and ii) those who focus on traditional LfD for manipulation to achieve extreme data efficiency with low parameter models (but are either limited to highly constrained problems or depend on expertly designed task spaces and operational space controllers).
>     - **Our work attempts to combine the best of both these areas in order to learn more complex dexterous manipulation behaviors with larger models (compared to traditional LfD for manipulation) and show that models with the inductive biases of Geometric Fabrics offer improved data efficiency and theoretical guarantees (compared to deep learning methods).**
>
> 3. It is a factual error to state that NGF is not a high-dimensional policy.
>     - **The NGF has full control authority over all 23 joints.**
>
> For more details, please refer to the top-level comments and responses to each individual reviewer.